# Assessing Performance of the Veterans Affairs Women Cardiovascular Risk Model in Predicting a Short-Term Risk of Cardiovascular Disease Incidence Using United States Veterans Affairs COVID-19 Shared Data

**DOI:** 10.3390/ijerph181910005

**Published:** 2021-09-23

**Authors:** Haekyung Jeon-Slaughter, Xiaofei Chen, Bala Ramanan, Shirling Tsai

**Affiliations:** 1VA North Texas Health Care System, Dallas, TX 75216, USA; Bala.Ramanan@utsouthwestern.edu (B.R.); Shirling.Tsai@utsouthestern.edu (S.T.); 2Department of Internal Medicine, University of Texas Southwestern Medical Center, Dallas, TX 75390, USA; 3Sanofi, Bridgewater, NJ 08807, USA; xiaofeic@mail.smu.edu; 4Department of Surgery, University of Texas Southwestern Medical Center, Dallas, TX 75390, USA

**Keywords:** women’s heart disease, women veterans, cardiovascular disease, cardiovascular risk score, COVID-19 and heart disease, short-term risk of heart disease with COVID-19

## Abstract

The current study assessed performance of the new Veterans Affairs (VA) women cardiovascular disease (CVD) risk score in predicting women veterans’ 60-day CVD event risk using VA COVID-19 shared cohort data. The study data included 17,264 women veterans—9658 White, 6088 African American, and 1518 Hispanic women veterans—ever treated at US VA hospitals and clinics between 24 February and 25 November 2020. The VA women CVD risk score discriminated patients with CVD events at 60 days from those without CVD events with accuracy (area under the curve) of 78%, 50%, and 83% for White, African American, and Hispanic women veterans, respectively. The VA women CVD risk score itself showed good accuracy in predicting CVD events at 60 days for White and Hispanic women veterans, while it performed poorly for African American women veterans. The future studies are needed to identify non-traditional factors and biomarkers associated with increased CVD risk specific to African American women and incorporate them to the CVD risk assessment.

## 1. Introduction

With the on-going COVID-19 pandemic continuing over 1 year, the impact of COVID-19 on health outcomes is both direct and indirect. Heart disease is one of many chronic health problems that were greatly affected during the COVID-19 pandemic, and death rates from heart disease increased during the pandemic [1]. Having pre-existing cardiovascular disease (CVD) before contracting COVID-19 was significantly associated with mortality [2,3].

Based on analysis of the US Veterans Affairs (VA) COVID-19 Shared Resources database, Tsai and colleagues [2] reported that women with COVID-19 infection showed a higher risk of myocardial injury (high-sensitivity cardiac troponin ≥0.4 mg/dL) than those without COVID-19. Among women with COVID-19 infection, pre-existing CVD was positively associated with an elevated risk of a CVD event occurrence within 60 days. These results suggest a need to assess women’s short term CVD risk during the ongoing COVID-19 pandemic.

The goal of this investigation is to assess women’s short-term CVD risk, and in particular, the CVD risk in young and minority groups, who are under-diagnosed and undertreated for CVD.

Traditional risk factors for CVD are older age, high blood pressure, high cholesterol, diabetes, and current smoking status. In addition, major depression is a known associated risk factor for CVD. The current widely used American College of Cariology/American Heart Association (ACC/AHA) Atherosclerosis Cardiovascular Disease (ASCVD) risk score [4] is comprised of traditional CVD risk factors; however, we have previously demonstrated that this risk score is not adequate to assess CVD risk for the young women under 40 years old and with a Hispanic background [5]. On the contrary, the CVD risk score developed by Jeon-Slaughter and colleagues [4] for Veterans Affairs (VA) women is acceptable for these groups of women. The new VA women CVD risk score was derived and internally validated from a development cohort of Non-Hispanic White, Non-Hispanic African American, and Hispanic women veterans aged between 30 and 79 [6].

The current study assessed the performance of the VA women CVD risk score in predicting short-term 60-day CVD events occurrence using the VA COVID-19 Shared Resources data, a dataset that is distinct from the derivation cohort used to develop the novel risk score.

## 2. Materials and Methods

### 2.1. A Study Cohort

The study used US Veterans Affairs (VA) COVID-19 shared data collected from 24 February to 25 November 2020, including 17,264 women veterans—9658 White, 6088 African American, and 1518 Hispanic women veterans—ever treated at US Veterans Affairs hospital clinics. This included women who tested positive and negative for SARS-CoV2. The VA COVID-19 shared data are extracted from VA electronic health records (EHR) [1]. and is a part of VA Infrastructure and Computing Infrastructure (VINCI) Corporate Data Warehouse (CDW) and available to VA researchers. The VA COVID-19 shared data includes patient-level data on COVID-19 testing, diagnoses at the time of or prior to COVID-19 testing, drug utilization, and 60-day follow-up after the incident COVID-19 test, including adverse events such as death and cardiovascular disease events. COVID-19 testing available in VA health care system are antigen, molecular including nucleic acid amplification; RNA, PCR, and antibody tests; and both confirmed and presumptive positive results were classified into positive test results [2].

### 2.2. VA Women CVD Risk Score and Its Calculation

The VA women CVD risk score was calculated stratified by race and ethnicity and was a predicted score in the percentage estimated at values of CVD risk factors—age, systolic blood pressure, current smoking status, presence of diabetes, major depression within 6 months, total cholesterol, and high-density lipoprotein-cholesterol (HDL-C)—at index date [6].

The CVD risk output from the VA women CVD risk score ranged between 0% and 100%. Each individual CVD risk score was calculated based on measured values of CVD risk factors at index date. The ACC/AHA guideline classified CVD risk score < 7.5% as a low risk, 7.5–19.9% as a moderate risk, and >20% as a high risk [4,5].

### 2.3. Statistical Methods

Means and standard deviations for continuous variables and frequencies and percentages for categorical variables were presented. Two-sided Wilcoxon rank test was used to examine the difference in VA women CVD risk score between those with and without CVD events in the 60 days post COVID-19 test. Cardiovascular disease events included ischemic and hemorrhagic stroke, Myocardial Injury (high-sensitivity cardiac troponin at 7 days > 0.4 ng/mL), heart failure, and arrythmia. Mantel-Haenszel Chi-Square test was conducted to examine associations of categorical covariates with experiencing any CVD events within 60 days.

Logistic regression was used to further test CVD risk score’s predictability of CVD events within 60 days post COVID-19 test after adjusting for confounders, stratified by race and ethnic groups. Confounders included in the models are women veterans’ age and BMI (Body Mass Index) at the time of COVID-19 test and the COVID-19 test result—positive vs. negative for SARS-CoV-2 infection. Odds Ratio (OR) and the corresponding 95% Confidence Interval (CI) were reported as logistic results. Receiver Operating Characteristics (ROC) Curves and Area under the curve (AUC) were used to examine accuracy of the model in predicting 60-day cardiovascular events among women veterans. An AUC of 70% and higher was considered as acceptable accuracy of the model. Model specification was tested and decided following Akaike Information Criteria (AIC) and significance of the included covariates.

A sample size was estimated based on number of events required per predictor (EPP) in the model. The minimum required EPP is 5 to 10 [7]. A minimum 15 to 30 CVD events per race/ethnic group are needed for a valid predictive model.

All statistical and graphic analyses were conducted using SAS 9.4 version (SAS Institute, Cary, NC, USA).

## 3. Results

Overall, 5.5% (n = 958) of the study cohort had experienced any cardiovascular disease events within 60 days after a reference time point identified as the time of COVID-19 testing. Analysis based on race and ethnicity revealed that CVD events occurred in 5.7% (n = 552) of White women veterans, 6.0% (n = 566) of African American women veterans, and 2.6% (n = 40) of Hispanic women veterans.

Women veterans in the study cohort who experienced CVD events were older, more likely to present with complications such as chronic kidney disease and COPD than those without CVD events (Table 1).

Women veterans with 60-day CVD events also had significantly higher Systolic Blood Pressure (SBP), higher rates of being prescribed antihypertensive medication, carrying a diagnosis of diabetes, and being current smokers, and lower HDL-C and total cholesterol levels than those who did not suffer a CVD event. There was no statistical difference found in major depression between women veterans with and without CVD events in 60 days after the reference time point (Table 1).

Interestingly, the number of CVD events within 60 days was significantly higher among women veterans with negative COVID-19 test results (8.5%) than women with positive COVID-19 test results (6.6%, *p* < 0.001). Testing positive for SARS-CoV-2 infection was not significantly associated with any CVD event occurrence within 60 days, even when stratified by three racial and ethnic groups.

The VA women CVD risk score at the reference time point was significantly higher in those who subsequently had CVD events in the following 60 days (mean 24% with standard deviation of 24.8%) than those who did not (mean 13% with standard deviation of 18%).

Figure 1 showed that CVD risk score alone discriminated patients with CVD events within 60 days from those who did not with accuracy (Area Under the Curve) of 78%, 50%, and 74% for White, African American, and Hispanic women veterans, respectively. An adjusted model, including COVID-19 test results and BMI in addition to CVD risk score as covariates, discriminated patients with CVD events within 60 days from those who did not with accuracy (area under the curve) of 81%, 75%, and 83% for White, African American, and Hispanic women veterans, respectively.

A body mass index (BMI, kg/m^2^) greater than 30 was significantly associated with increased risk of 60-day CVD events across three race and ethnic groups (White OR 1.09, 95% CI 1.05–1.14; African American OR 1.11 95% CI 1.06–1.16; Hispanic OR 1.20, 95% CI 1.02–1.38).

White women veterans faced an 8.5-fold increased risk of having CVD events within 60 days for each one-percent increase in CVD risk score, while the CVD risk score was not significantly associated with 60-day CVD events among African American and Hispanic women veterans.

A positive COVID-19 test was not significantly associated with an increased risk of CVD events within 60 days across all three race and ethnic groups.

## 4. Discussion

Young and minority women are under-diagnosed and undertreated for CVD due to a lack of an acceptable CVD risk assessment tool. The current recommended treatment guideline for high cholesterol and high blood pressure [8,9] are built upon the CVD risk assessment; thus, using a risk score that adequately assesses CVD risk is crucial to provide optimal CVD treatment for women. The study tested the performance of a new VA women CVD risk score, which is tailored to women veterans and adequate for young women under 40 years old and with Hispanic background, in assessing short-term CVD risk. The VA women CVD risk score can be used to assess CVD risk for women without known CVD history—in particular, young and Hispanic women who are often underdiagnosed for CVD.

The VA women CVD risk score alone predicted 60-day CVD events with an accuracy of 78% and 77% for non-Hispanic White and Hispanic women veterans, respectively. On the contrary, traditional CVD risk factors and major depression alone failed to predict a short-term CVD event risk for African American women veterans (<50% accuracy). However, with additional factors such as BMI and COVID-19 infection, the prediction accuracy for African American women was improved to 75%.

The poor accuracy of VA women CVD risk score in African American women may be due to a fact that many African American women in current study data (23%) had a history of CVD events, while the VA women CVD risk score was designed for those without a history of CVD events. The model accuracy for African American women veterans’ 60-day CVD event risk during COVID-19 pandemic was improved with inclusion of other risk factors—BMI and COVID-19 infection status—in addition to those already included in the VA women CVD risk score. BMI and COVID-19 infection may serve as proxy variables measuring confounding effects of cultural and socio-economic factors on increased risk of CVD event occurrence. African American women veterans had higher rates of obesity and COVID-19 infection than White women veterans (54% vs. 45%, *p* <0.0001), which partly accounts higher CVD-related death in African Americans during COVID-19 pandemic than their White counterparts [10]. Additionally, the prevalence of pregnancy complications such as preeclampsia, a known risk factor for increased CVD risk, which was not included in the current model, among African American women is significantly higher than White and other race women [11,12].

In Hispanic women veterans, VA women CVD risk score predicted CVD events with more than 80% accuracy, but the score was not significantly associated with CVD events at 60 days due to a small sample size.

The study has many limitations. The length of follow-up was 60 days given the available data, and a long-term follow-up is needed. The current data may have limitations in validating the VA women CVD risk score due to its short-term follow-up. A future study with long-term follow-up data is warranted. The study did not include any biomarkers in the prediction model due to a high rate of missing data. The VA COIVD-19 Shared Resources data did not include sex-specific conditions, such as a history of pregnancy complications, which is known to be associated with elevated CVD risk. The current study data were extracted from electronic medical records and heavily relied on International Classification of Disease (ICD) diagnosis codes. Thus, we cannot exclude a possibility of human error in the medical record entry.

## 5. Conclusions

The VA women CVD risk score itself showed a good accuracy in predicting CVD events within 60 days after a reference timepoint for White and Hispanic women veterans, but not for African American women. This is likely due to omittance bias, wherein the traditional CVD risk factors alone did not predict short-term CVD event risk for African American women veterans. Future studies are needed to identify non-traditional factors and biomarkers that are associated with a short-term CVD risk among African American women veterans. The VA women CVD risk score can serve as a cardiovascular point-of-care tool for women patients without a known history of CVD events or complications.

## Figures and Tables

**Figure 1 ijerph-18-10005-f001:**
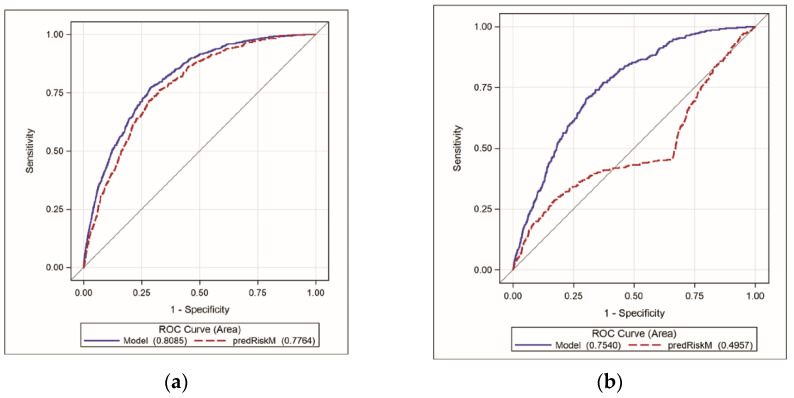
Receiver operating characteristics (ROC) curves of the model and CVD risk score for three race and ethnic groups. (**a**) White women veterans; (**b**) African American women veterans; (**c**) Hispanic women veterans. Notes: blue lines represent ROC curves for the model and red dashed lines for CVD risk score. Numbers in parentheses represent areas under the ROC curves. A 45-degree line represents 50% accuracy.

**Table 1 ijerph-18-10005-t001:** Baseline characteristics of veteran women by cardiovascular disease (CVD) event 60 days after COVID-19 testing.

Baseline Characteristics at Index			All (n = 17,264)	CVD Event (n = 958)	No CVD Event (n = 16,306)	*p*-Value ^a^
Age at index	Years	mean ± SD	52.48 ± 12.59	63.33 ± 9.03	51.84 ± 12.47	<0.0001
BMI at index	kg/m^2^	mean ± SD	31.00 ± 7.20	32.22 ± 8.57	30.93 ± 7.10	<0.0001
Race	White	n (%)	9658 (55.94)	552 (5.72)	9.106 (94.3)	<0.0001
	African American	n (%)	6611 (35.26)	366 (6.01)	5722 (93.99)	
	Hispanics	n (%)	1518 (8.79)	40 (2.64)	1478 (97.36)	
Systolic Blood pressure at index	mm Hg	mean ± SD	130.87 ± 20.12	134.7 ± 25.78	130.72 ± 19.73	<0.0001
Total cholesterol	mg/dL	mean ± SD	190.00 ± 43.51	175.74 ± 50.90	190.83 ± 42.89	<0.0001
HDL-C	mg/dL	mean ± SD	55.54 ± 17.42	52.05 ± 18.18	55.74 ± 17.36	<0.0001
On Antihypertensive medication	Yes	n (%)	5401 (31.28)	519 (54.18)	4882 (29.94)	<0.0001
DM	Yes	n (%)	4301 (24.91)	536 (55.94)	3765 (23.09)	<0.0001
Current smoker	Yes	n (%)	3238 (18.76)	226 (23.56)	3012 (18.47)	<0.0001
Current major depression (<6 months)	Yes	n (%)	3428 (19.86)	183 (19.10)	3245 (19.90)	0.5472
CVD risk score ^b^	%	mean ± SD	13.31 ± 19.27	23.65 ± 24.80	12.70 ± 18.72	<0.0001
COVID-19 testing result	Positive (+)	n (%)	1445 (8.37)	63 (6.58)	1382 (8.48)	0.0391

Abbreviations. BMI = body mass index; CVD = cardiovascular disease; DM = diabetes; HDL-C = high-density lipoprotein-cholesterol. ^a^ T- and Chi-squared tests were used for continuous and categorical characteristics, respectively. ^b^ VA women CVD risk score (Jeon-Slaughter et al., 2021 [6]).

## Data Availability

Due to the sensitive nature of the data collected for this study, requests to access the dataset are limited to qualified VA-affiliated researchers trained in human subject confidentiality. Protocols may be sent to the VA North Texas Health Care System IRB at NTXIRBAdmin@va.gov. SQL, SAS, and R code used in the analysis of this study are available from the corresponding author upon reasonable request. All methods were performed in accordance with the relevant guidelines and regulations.

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
