# Peer review of "Assessing Performance of the Veterans Affairs Women Cardiovascular Risk Model in Predicting a Short-Term Risk of Cardiovascular Disease Incidence Using United States Veterans Affairs COVID-19 Shared Data"

_ijerph, 2021, doi:10.3390/ijerph181910005_

Round 1
Reviewer 1 Report
In the introduction, it is important to make clear that this scoring system is for all women but traditionally young women and Hispanic women were under diagnosed.
Also, it needs to be made clearer that the thought was that COVID19 itself did not increase the risk of cardiovascular disease. Clearly spell out the most common risk factors, and that those were associated with the greatest risk of events, not COVID.
Add key words- COVID19 and heart disease, short term risk of heart disease with COVID
In the discussion and conclusion- There needs to be clarity surrounding the actual risk score and that for African American women, the risk factors need to include BMI and perhaps complications of pregnancy.
It is unclear in stating that COVID19 positivity is not associated with an increase in cardiovascular risk, but is that not true for African American women? It states that adding the COVID test and BMI might increase the accuracy for African American women. How does the role of the COVID test increase accuracy.
You allude to the 60 day risk. Is this a long enough period of time to determine true impact of these risk factors on outcomes?
Author Response
We thank you for insightful and helpful comments. We extensively revised introduction and discussion in response to reviewers’ comment. We corrected references cited.
Reviewer 1:
In the introduction, it is important to make clear that this scoring system is for all women but traditionally young women and Hispanic women were under diagnosed.
Response: Thank you for your comments we extensively revised introduction in response. The revised manuscript clearly states this.
Also, it needs to be made clearer that the thought was that COVID19 itself did not increase the risk of cardiovascular disease. Clearly spell out the most common risk factors, and that those were associated with the greatest risk of events, not COVID.
Response: Thank you for your comments we added sentences explain traditional risk factors associated with elevated cardiovascular risk.
Add key words- COVID19 and heart disease, short term risk of heart disease with COVID
Response: Suggested key words are added.
In the discussion and conclusion- There needs to be clarity surrounding the actual risk score and that for African American women, the risk factors need to include BMI and perhaps complications of pregnancy.
Response: Thank you for your comments we revised discussion in lines 171-185.
It is unclear in stating that COVID19 positivity is not associated with an increase in cardiovascular risk, but is that not true for African American women? It states that adding the COVID test and BMI might increase the accuracy for African American women. How does the role of the COVID test increase accuracy.
Response: Thank you for your comments, in response we added “The poor accuracy of VA women CVD risk score in African American women…. on women veterans.[4]” as a possible explanations in lines 170-194.
You allude to the 60 day risk. Is this a long enough period of time to determine true impact of these risk factors on outcomes?
Response: We agree with reviewer's concern. Using covid-19 data restricted to us to examine only a short term and a full external validation requires a 5-10 year follow up. In addition, we fully expanded this as a limitation (lines 199-200). Despite this limitation, VA CVD women risk score has a utility for covid positive patients if she is underdiagnosed for CVD, when CVD preexisting condition is associated with subsequent CVD occurrence.
Reviewer 2 Report
Dr. Shirling Tsai and her colleagues evaluate the performance of the Veterans Affairs women cardiovascular risk model in predicting 60-days risk of cardiovascular disease incidence during Covid-19 Pandemic by using United States Veterans Affairs COVID-19 Shared Data.
In this study, the researcher focused on women and included ethnicity/race in their evaluation. I greatly respect their efforts as we have limited clinical information from the minorities and these kinds of studies help us improve health disparity among women, especially in minorities.
The scientific facts in this paper do not match with the references.
1- Line 27-30
With on-going COVID-19 Pandemic for over 1 year, the impact of COVID-19 on health outcomes is both direct and indirect. Heart disease is one of many that were greatly affected during the COVID-19 Pandemic, and death rates from heart disease increased during the Pandemic.[1]
NOT MATHC WITH REF 1
1- Compton WM, Thomas YF, Stinson FS, Grant BF. Prevalence, correlates, disability, and comorbidity of DSM-IV drug abuse 240 and dependence in the United States: results from the national epidemiologic survey on alcohol and related conditions. Archives of 241 general psychiatry. 2007;64(5):566-576
2- Line 30-31
Having pre-existing cardiovascular disease (CVD) before con- tracting COVID-19 was significantly associated with mortality,[2,3]
NOT MATHC WITH REF 2 and 3
- Center for Behavioral Health Statistics and Quality. Behavioral health trends in the United States: Results from the 2014 National Survey on Drug Use and Health (In: HHS, ed. Vol No. SMA 15-4927. Rockville, MD: HHS Publication; 2015
3- Hernandez-Avila CA, Rounsaville BJ, Kranzler HR. Opioid-, cannabis- and alcohol-dependent women show more rapid progression to substance abuse treatment. Drug Alcohol Depend. 2004;74(3):265-272
3- Line 35-38
A new CVD risk score targeted for women veterans aged between 30 and 79— the Veterans Affairs (VA) women CVD risk score— was developed and internally validated using a development cohort of women veterans’ electronic health records between 2007 and 2017.[4]
NOT MATHC WITH REF 4
- Kosten TA, Gawin FH, Kosten TR, Rounsaville BJ. Gender differences in cocaine use and treatment response. Journal of substance abuse treatment. 1993;10(1):63-66.
4- Line 38-41
The new VA women CVD risk score better performed in predicting 10-year
CVD risk for women veterans than the widely used the American College of Cariology/American Heart Association (ACC/AHA) Atherosclerosis Cardiovascular Disease (ASCVD) risk score.[5]
NOT MATHC WITH REF 5
- Greenfield SF, Back SE, Lawson K, Brady KT. Substance Abuse in Women. The Psychiatric clinics of North America.2010;33(2):339-355.
5- Line 41-42
In particular, the ACC/AHA ASCVD risk score is not accurate for predicting CVD risk in younger and Hispanic women,[6]
NOT MATHC WITH REF 6
- NIH NIDA Research Reports: Substance use in Women Publisjed in September 2016 available at https://www.drugabuse.gov/publications/research-reports/substance-use-in-women/summary accessed on 06/13/2018
6- Line 87
Delete extra dot after results.
87 dence Interval (CI) were reported as logistic results.
7- Line 154-155
There were excess deaths from underlying heart disease during 2020 (5% increase in annual percentage change),[7].
NOT MATHC WITH REF 7
- Benedetti F. Placebo Effects: From the Neurobiological Paradigm to Translational Implications. Neuron. 2014;84(3):623-637
8- Line 155-158
This may have been caused by delayed and deferred care during COVID-19 Pandemic and was more prominent among minority groups during the COVID-19 Pandemic.[8]
NOT MATHC WITH REF 8
- Polak K, Haug NA, Drachenberg HE, Svikis DS. Gender Considerations in Addiction: Implications for Treatment. Curr Treat Options Psychiatry. 2015;2(3):326-338
9- Line 56
This included women who tested positive and negative for SARS-coV2.
Would you please specify the method of testing? Was it PCR or Rapid test?
10-
It is not clear what is the central hypothesis of this study. Based on the study subject, it seems that the hypothesis was the validity of VA CVD score to predict CVD events in 60-days. Still, on the paper, the authors discuss a lot about Covid-19 and cardiovascular events and based on their findings, the CVD events were less in patients with positive Covid-19.
If the main idea was evaluating the usefulness of VA CVD score during the Covid-19 Pandemic, the United States Veterans Affairs COVID-19 Shared Data is not a suitable database for this goal. Based on the findings of this study, using the VA CVD score is a fair helpful tool to evaluate CVD events risk in subjects that have Covid-19 test reports, and the selected sample is biased for applying these findings to the general population. The discussion and conclusion should be amended accordingly.
Author Response
We thank you for insightful and helpful comments. We extensively revised introduction and discussion in response to reviewers’ comment. We corrected references cited.
Dr. Shirling Tsai and her colleagues evaluate the performance of the Veterans Affairs women cardiovascular risk model in predicting 60-days risk of cardiovascular disease incidence during Covid-19 Pandemic by using United States Veterans Affairs COVID-19 Shared Data.
In this study, the researcher focused on women and included ethnicity/race in their evaluation. I greatly respect their efforts as we have limited clinical information from the minorities and these kinds of studies help us improve health disparity among women, especially in minorities.
Response: In response, we fixed references cited.
The scientific facts in this paper do not match with the references.
1- Line 27-30
With on-going COVID-19 Pandemic for over 1 year, the impact of COVID-19 on health outcomes is both direct and indirect. Heart disease is one of many that were greatly affected during the COVID-19 Pandemic, and death rates from heart disease increased during the Pandemic.[1]
NOT MATHC WITH REF 1
1- Compton WM, Thomas YF, Stinson FS, Grant BF. Prevalence, correlates, disability, and comorbidity of DSM-IV drug abuse 240 and dependence in the United States: results from the national epidemiologic survey on alcohol and related conditions. Archives of 241 general psychiatry. 2007;64(5):566-576
2- Line 30-31
Having pre-existing cardiovascular disease (CVD) before con- tracting COVID-19 was significantly associated with mortality,[2,3]
NOT MATHC WITH REF 2 and 3
- Center for Behavioral Health Statistics and Quality. Behavioral health trends in the United States: Results from the 2014 National Survey on Drug Use and Health (In: HHS, ed. Vol No. SMA 15-4927. Rockville, MD: HHS Publication; 2015
3- Hernandez-Avila CA, Rounsaville BJ, Kranzler HR. Opioid-, cannabis- and alcohol-dependent women show more rapid progression to substance abuse treatment. Drug Alcohol Depend. 2004;74(3):265-272
3- Line 35-38
A new CVD risk score targeted for women veterans aged between 30 and 79— the Veterans Affairs (VA) women CVD risk score— was developed and internally validated using a development cohort of women veterans’ electronic health records between 2007 and 2017.[4]
NOT MATHC WITH REF 4
- Kosten TA, Gawin FH, Kosten TR, Rounsaville BJ. Gender differences in cocaine use and treatment response. Journal of substance abuse treatment. 1993;10(1):63-66.
4- Line 38-41
The new VA women CVD risk score better performed in predicting 10-year
CVD risk for women veterans than the widely used the American College of Cariology/American Heart Association (ACC/AHA) Atherosclerosis Cardiovascular Disease (ASCVD) risk score.[5]
NOT MATHC WITH REF 5
- Greenfield SF, Back SE, Lawson K, Brady KT. Substance Abuse in Women. The Psychiatric clinics of North America.2010;33(2):339-355.
5- Line 41-42
In particular, the ACC/AHA ASCVD risk score is not accurate for predicting CVD risk in younger and Hispanic women,[6]
NOT MATHC WITH REF 6
- NIH NIDA Research Reports: Substance use in Women Publisjed in September 2016 available at https://www.drugabuse.gov/publications/research-reports/substance-use-in-women/summary accessed on 06/13/2018
6- Line 87
Delete extra dot after results.
87 dence Interval (CI) were reported as logistic results.
7- Line 154-155
There were excess deaths from underlying heart disease during 2020 (5% increase in annual percentage change),[7].
NOT MATHC WITH REF 7
- Benedetti F. Placebo Effects: From the Neurobiological Paradigm to Translational Implications. Neuron. 2014;84(3):623-637
8- Line 155-158
This may have been caused by delayed and deferred care during COVID-19 Pandemic and was more prominent among minority groups during the COVID-19 Pandemic.[8]
NOT MATHC WITH REF 8
- Polak K, Haug NA, Drachenberg HE, Svikis DS. Gender Considerations in Addiction: Implications for Treatment. Curr Treat Options Psychiatry. 2015;2(3):326-338
9- Line 56
This included women who tested positive and negative for SARS-coV2.
Would you please specify the method of testing? Was it PCR or Rapid test?
Response: In response, we added “COVID testing available in VA health care system are antigen, molecular including nucleic acid amplifiction, RNA, and PCR, and antibody tests, and both confirmed and presumptive positive results were classified into positive test results.[2]” in lines 64-66.
10-
It is not clear what is the central hypothesis of this study. Based on the study subject, it seems that the hypothesis was the validity of VA CVD score to predict CVD events in 60-days. Still, on the paper, the authors discuss a lot about Covid-19 and cardiovascular events and based on their findings, the CVD events were less in patients with positive Covid-19.
Response: Thank you for your feedback. We extensively revised our discussion to clarify. The revised discussion focuses on the performance test of VA women CVD risk score in predicting short term CVD events using VA COVID-19 resources data.
If the main idea was evaluating the usefulness of VA CVD score during the Covid-19 Pandemic, the United States Veterans Affairs COVID-19 Shared Data is not a suitable database for this goal. Based on the findings of this study, using the VA CVD score is a fair helpful tool to evaluate CVD events risk in subjects that have Covid-19 test reports, and the selected sample is biased for applying these findings to the general population. The discussion and conclusion should be amended accordingly.
Response: We agreed with the reviewer in VA COVID-19 shared data may not be an ideal external validation cohort due to a short-term duration. The VA women CVD risk score derived from women veterans data may need recalibration to apply to the general population. This was added in the revised discussion as limitation.
Round 2
Reviewer 2 Report
Dr. Haekyung Jeon-Slaughter and her colleagues evaluate the Veterans Affairs women's cardiovascular risk model's performance in predicting 60-days risk of cardiovascular disease incidence during Covid-19 Pandemic using United States Veterans Affairs COVID-19 Shared Data.
I appreciate the efforts of the research team to address my concern, but my concerns about the study design have not been addressed.
This study assesses the performance of the validated tool (an Internally Validated Veterans Affairs Women Cardiovascular Disease Risk Score Using Veterans Affairs National Electronic Health Records) to predict the short-term risk of cardiovascular disease incidence by using United States Veterans Affairs COVID-19 Shared Data.
1- The published study by Dr. Haekyung Jeon-Slaughter in JAHA on Feb 10, 2021 (Jeon-Slaughter H, Chen X, Tsai S, Ramanan B, Ebrahimi R. Developing an Internally Validated Veterans Affairs Women Cardiovascular Disease Risk Score Using Veterans Affairs National Electronic Health Records. J Am Heart Assoc. 2021 Feb;10(5):e019217. doi: 10.1161/JAHA.120.019217. Epub 2021 Feb 23. PMID: 33619994; PMCID: PMC8174271.) showed the validity of this tool to predict 10-year CVD risk.
If the aim of this study is to assess the Veterans Affairs Women Cardiovascular Disease Risk Score for predicting short-term risk of cardiovascular disease, why did the researcher not use Veterans Affairs National Electronic Health Records for this purpose? What is the justification for changing the database? Why did Dr. Haekyung Jeon-Slaughter and her team decide not to analyze this tool's accuracy to predict the short-term risk of cardiovascular disease on their JAHA paper and publish this study as a separate study?
If the aim of this study is assessing the Veterans Affairs Women Cardiovascular Disease Risk Score for predicting short-term risk of cardiovascular disease in Covid-19 patients, the analysis should be amended accordingly. These analyses should be done among the subjects with Covid-19 positive test from United States Veterans Affairs COVID-19 Shared Data.
2- Line 371-372
Conclusion:
The VA women CVD risk score itself showed a good accuracy in predicting CVD events within 60 days after COVID testing.
The published study in JAHA showed the validity of this tool for predicting 10-year CVD risk among White, Black, and Hispanic women service members and veterans aged 30 to 79 years. What is the justification of the difference between patients with the COVID-19 test and other treated patients in the VA system? What is the unique property of patients with the Covid-19 test that the researcher decided to test the accuracy of this tool among patients who had a history of the Covid-19 test? Having the Covid-19 test is not a confounder nor an effect modifier that could change the validity/accuracy of the Veterans Affairs Women Cardiovascular Disease Risk Score.
In a not shell,
A) If the hypothesis of this study is assessing the accuracy of Veterans Affairs Women Cardiovascular Disease Risk Score for predicting short-term risk of cardiovascular disease, it is better the researcher use Veterans Affairs National Electronic Health Records like an original study that was published in JAHA in Feb 2021 and also determine the justification of trying to publish two different studies rather than doing the short- prediction accuracy in the original research.
B) If the hypothesis of this study is assessing the accuracy of Veterans Affairs Women Cardiovascular Disease Risk Score for predicting short-term risk of cardiovascular disease during Covid-19 pandemic, the study should compare the accuracy of this risk score among the Covid-19 positive and negative subjects and then conclude if this is a proper tool or not during Covid-19 pandemic.
Author Response
Thank you for your feedback in response, we revised the manuscript adding explanation of the study data to clarify to the adequacy of the data for the study objective and deleted discussion rather related to the study data than the study’s objective.
Please see the more details below.
On reviewer #2’s comment on a study design, we believe the current study design is adequate for a study’s objective: the study considered the following statistical methods and conceptual designs to decide to use the VA COVID-19 registry data without stratifying by COVID-19 test results (positive vs negative): All of these were addressed in the revised manuscript.
1) External validation data has to be different from the original development cohort: Thus we can not use the same development cohort data (Janssen KJ, Moons KG, Kalkman CJ, Grobbee DE, Vergouwe Y. Updating methods improved the performance of a clinical prediction model in new patients. J Clin Epidemiol. Jan 2008;61(1):76-86) We added the “, a dataset that is distinct from the derivation cohort used to develop the novel risk score.” In line 57-58 for clarification.
2) Stratifying by covid-19 test results will result in model overfitting problems--due to a relatively small number of CVD events among COVID-19 test positive patients (n=63) following a sample size estimation rule in predictive modeling. This was estimated based on number of events required per predictor (EPP) in the risk score model. The minimum required EPP is 5 to 10.( Vittinghoff E, McCulloch CE. Relaxing the rule of ten events per variable in logistic and Cox regression. American journal of epidemiology. 2007;165(6):710-718.) A minimum 15 to 30 CVD events per race/ethnic group are needed for a valid predictive model. This was added to the method section line 97-99 with a new reference 6 (replaced the previous 6).
3) The study assesses utility/performance of the VA women CVD risk score for cardiovascular care at health care settings during the COVID-19 pandemic era. To achieve this goal and to validate the VA women CVD risk score as a clinical decision tool for cardiovascular care at a health care setting, it needs to be validated using data including both positive and negative COVID-19 patients. The tool will be used to assess CVD risk simultaneously with COVID-19 test. One hand, as having a CVD history is a predictor of COVID-19 severity after hospitalization due to COVID-19 so it is essential to assess CVD risk at admission when women are underdiagnosed with CVD. On the other hand, the most recent U.S. excess mortality data showed that heart disease related mortality is a leading cause of non-covid-19-related mortality and it affected women disproportionately in increased percent of deaths in young and middle aged women despite absolute numbers of deaths are still higher among men than women.
Indirectly, COVID-19 affected heart-related mortality during COVID-19 pandemic among women with negative COVID-19 test results as well (increased excess deaths related heart disease).